# Torque and Battery Distribution Strategy for Saving Energy of an Electric Vehicle with Three Traction Motors

**Yi-Hsiang Tseng [1] and Yee-Pien Yang [1,2,*]**

1   Department of Mechanical Engineering, National Taiwan University, Taipei 10617, Taiwan;
    gn0000023@gmail.com

2   Mechanical and Mechatronics Systems Research Laboratories, Industrial Technology Research Institute,
    Hsinchu 31057, Taiwan

*   Correspondence: ypyang@ntu.edu.tw or yeeyang@itri.org.tw; Tel.: +886-2-3366-2682

**Abstract:** A torque and battery distribution (TBD) strategy is proposed for saving energy for an electric vehicle (EV) that is driven by three traction motors. Each traction motor is driven by an independent inverter and a battery pack. When the vehicle is accelerating or cruising, its vehicle control unit determines the optimal torque distribution of the three motors by particle swarm optimization (PSO) theory to minimize energy consumption on the basis of their torque–speed–efficiency maps. Simultaneously, the states of charge (SOC) of the three battery packs are controlled in balance for improving the driving range and for avoiding unexpected battery depletion. The proposed TBD strategy can increase 7.7% driving range in the circular New European Driving Cycle (NEDC) of radius 100 m and 28% in the straight-line NEDC. All the battery energy can be effectively distributed and utilized for extending the driving range with an improved energy consumption efficiency.

**Keywords:** torque and battery distribution; particle swarm optimization; electric vehicle

## 1. Introduction

Hybrid and pure electric vehicles (EVs) have been commercially available for many years. A lot of research that focused on multiple propulsion and energy storage systems and the related power split and energy economy strategies has become imperative issue in EVs. For EVs with only one battery pack, it is important to balance the capacity of battery cells for improving its lifetime. Li et al. [1] presented a real-time state-of-charge (SOC) calculation method for a pure EV, where the lithium battery was simulated with a second-order resistance–capacitance (RC) model and the remaining capacity in battery cells was balanced by fuzzy control through a set of bi-directional fly-back direct current-direct current (DC–DC) converters. Gallardo-Lozano et al. [2] introduced a shunting transistor method to balance battery cells during the recharging and driving modes. Huang and Abu Qahouq [3] proposed an energy sharing control scheme to regulate the DC bus voltage, and simultaneously, to balance the SOC of battery cells with micro DC–DC converters. Pham et al. [4] addressed a fast-balancing topology for lithium-ion batteries in an EV by transferring the power in high-voltage cells directly to low-voltage cells through DC–DC converters. During battery charging, Dung et al. [5] eliminated the racing phenomenon by a pulse width modulation (PWM) based equalization process among battery packs so that the charging time was reduced by 48%.

For EVs with hybrid energy storage systems (HESSs), Jin et al. [6] and Akar et al. [7], respectively, proposed for their HESS of batteries and ultracapacitors a fuzzy control-based power management strategy to reduce battery degradation. A PWM technique was introduced by Menon et al. [8] for balancing the SOC of independent battery packs by continuously regulating the power flow from

two inverters on the basis of the driving demands. Tanaka et al. [9] used two batteries in a hybrid EV for investigating a high-efficiency energy conversion system to improve the driving range. The main battery provided fundamental power that did not need to be passed through a DC–DC converter, while additional power was supplied by a sub-battery through the DC–DC converter.

Lately, pure EVs with multiple traction motors have been commercially available. Examples are the Porsche Mission E Cross Turismo with two permanent magnet synchronous motors (PMSMs) and the Audi e-tron quattro with three traction motors. Rossi et al. [10] introduced a two-motor, two-axle, two-battery pack powertrain configuration for a compact EV and proposed an optimal front-rear motor transmission combination for the best driving performance. Several advantages of multiple motors and battery packs were addressed: the increased fault tolerance; the reduction of power rating in electric drive with possible simplification and cost reduction; the reduction of insulation level and electromagnetic emission of low-voltage power modules; and the additional degrees of freedom in torque vectoring for stable vehicle maneuverability.

Some studies have focused on the driving and braking torque distributions on motors for vehicle stability and handling performance. Yin at al. [11] used a hierarchical electronic stability controller (ESC) to distribute direct torque to four in-wheel motors of an EV for improving the vehicle stability and handling performance. Zhai et al. [12] proposed a similar ESC algorithm to improve vehicle stability by distributing the driving and regenerative braking torque for an EV with four independent in-wheel motors. Other studies have focused on the energy economy of EVs that use torque split strategies to arrange multiple traction motors. Dizqah et al. [13] formulated a parametric energy-efficient torque distribution optimization problem depending on the speed of an EV driven by four identical drivetrains, resulting in an energy consumption reduction of 0.1%–0.5% under various European driving cycles. An EV with four in-wheel motors was introduced by Fujimoto and Harada [14] where the slip ratio and motor loss were optimized on the basis of the vehicle speed and acceleration over the Japanese JC08 cycle. Sun et al. [15] proposed an online braking torque allocation scheme for a four-wheel-drive EV that minimized tire and electromechanical losses. Simulations showed that the driving efficiency was increased 4.3% in high speed driving cycles and 1.5% in normal speed driving cycles.

Yang et al. [16] proposed a real-time torque distribution strategy for a pure EV with three motors and three battery packs. The front wheels were driven indirectly by a traction motor through reduction gears, while two rear wheels were driven directly by two in-wheel motors. Torque distribution was determined by the particle swarm optimization (PSO) theory for minimizing energy consumption on the basis of the torque–speed–efficiency (TNE) maps of all the traction motors. Subsequently, Yang and Chen [17] introduced a coupled parallel energy saving and safety strategy that minimized energy consumption by torque distribution according to the PSO theory. The stabilizing direct yaw moment was also minimized on the basis of the stability region on the phase plane of sideslip angle and yaw rate.

Most of the above research focused either on the battery energy distribution to keep the battery cells in balance, or on the torque distribution for vehicle stability, handling, or energy economy. This paper extends the authors' previous study [18] that proposed a coupled parallel energy balancing and energy saving strategy by keeping the SOC of three independent battery packs in balance and distributing the driving torque of three traction motors during vehicle motion. Section 2 introduces vehicle configuration, longitudinal and lateral vehicle dynamics models, tire and transmission models, and battery SOC model. Section 3 elaborates the proposed torque and battery distribution strategy, and Section 4 provides experiments of model-in-the-loop (MIL) and hardware-in-the-loop (HIL) simulations, and road tests. Section 5 presents concluding remarks.

## 2. Vehicle Configuration

The EV was fitted with a 15-kW radial-flux PMSM that drove the two front wheels indirectly through a gearbox reducer and two identical 7-kW axial-flux PMSMs to drive the left and right rear wheels directly through the hubs (Figure 1a). The control strategy was validated by CarSim simulation

software with 15 mechanical degrees of freedom (DOF) for the four-wheeled vehicle. The steering system had one DOF, each wheel had one spin DOF, each suspension had two DOF, and the sprung mass was simplified as a rigid body with six DOF. The vehicle variables for the longitudinal and lateral dynamics models are defined in Figure 1b,c.

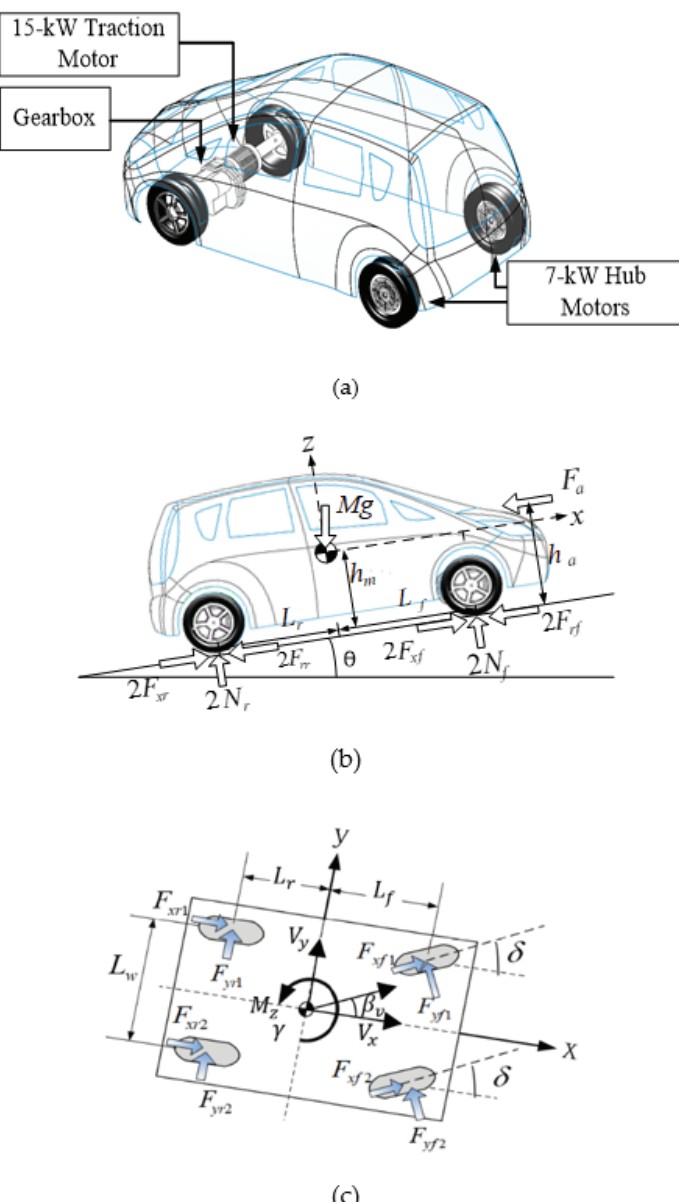

**Figure 1.** Electric vehicle configuration and variable definitions (**a**) propulsion system of multiple motors, and (**b**) the longitudinal and (**c**) lateral vehicle dynamics models.

Figure 2a,b provide the TNE maps from the experiments for the driving modes of the three traction motors. The braking modes were estimated from the mirror image of 75% efficiency of the driving mode. The maximum torque was 150 Nm and the maximum speed was 2400 rpm for the 15-kW front motor, and they were 122 Nm and 1200 rpm for the 7-kW rear motors. Three motor control units were responsible for driving the three motors, and three lithium-ion batteries were deployed for providing the power: one pack of 144 V, 72 Ah, and 10.45 kWh for the front drive and two packs of 72 V, 72 Ah, and 5.2 kWh for the two rear drives.

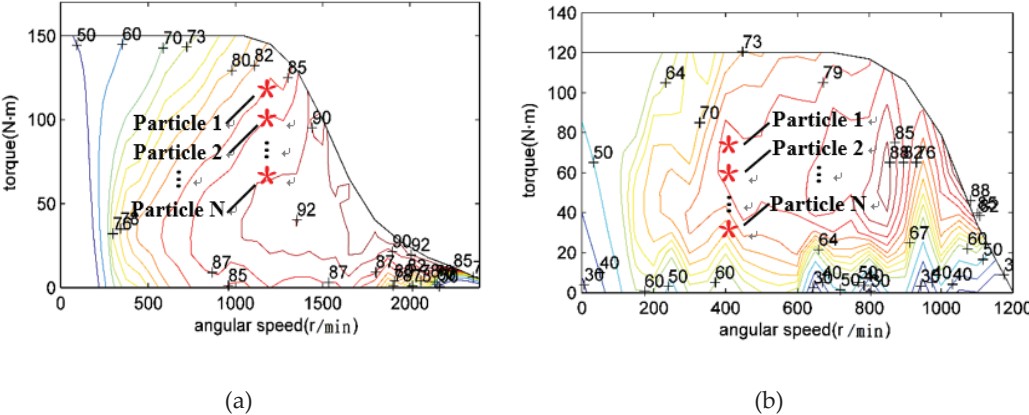

**Figure 2.** Torque-speed-efficiency maps: (**a**) the driving mode of the 15-kW front motor, (**b**) the driving modes of the two 7-kW rear motors.

### 2.1. Longitudinal Vehicle Dynamics Model

When the vehicle travels in a straight line, the longitudinal traction forces are usually simplified as $F_{xr} = F_{xr1} = F_{xr2}$ and $F_{xf} = F_{xf1} = F_{xf2}$. The force with subscript 1 is the force exerted on the left tire, while 2 on the right tire. The tractive force $F_x$ and the normal forces $N_f$ and $N_r$ of the front and rear wheels were obtained by the following equations:

$$F_x = M\dot{V}_x + Mg\sin\theta + \frac{1}{2}\rho C_d A_f V_x^2 + 2C_t\left(N_f + N_r\right) \tag{1}$$

$$N_f = \frac{M_v g L_r \cos\theta - \frac{1}{2}\rho C_d A_f V_x^2 h_a - \left[M_v h_m + 2\left(m_{wf} + m_{wr}\right)r_t\right]\dot{V}_x}{2\left(L_f + L_r\right)} + m_{wf}g\cos\theta \tag{2}$$

$$N_r = \frac{M_v g L_f \cos\theta + \frac{1}{2}\rho C_d A_f V_x^2 h_a + \left[M_v h_m + 2\left(m_{wf} + m_{wr}\right)r_t\right]\dot{V}_x}{2\left(L_f + L_r\right)} + m_{wr}g\cos\theta \tag{3}$$

$$F_x = 2\left(F_{xf} + F_{xr}\right) \tag{4}$$

$$M = M_v + 2\left(m_{wf} + m_{wr}\right) \tag{5}$$

where $M$ is the total vehicle mass; $M_v$ is the vehicle mass excluding tire and wheel mass; $m_{wf}$ is the front tire and wheel mass; $m_{wr}$ is the rear tire and wheel mass; $g$ is the gravity acceleration; $\theta$ is the slope angle in degrees; $\rho$ is the air density; $V_x$ is the longitudinal velocity of vehicle; $F_{xf}$ and $F_{xr}$ are the traction forces exerted on the front and rear tires; $L_f$ is the distance from mass center to front tire; $L_r$ is the distance from mass center to rear tire; and $r_t$ is the tire radius. Other vehicle specifications used in this paper are described in Table 1.

### 2.2. Lateral Vehicle Dynamics Model

As shown in Figure 1b, the lateral vehicle dynamics are described by a four-wheel model when the vehicle is cornering. The longitudinal, lateral, and yaw vehicle dynamic equations are expressed as:

$$\left(F_{xf1} + F_{xf2}\right)\cos\delta - \left(F_{yf1} + F_{yf2}\right)\sin\delta + F_{xr1} + F_{xr2} = M\left[\dot{V}_x - \left(\gamma + \frac{d\beta_v}{dt}\right)V_y\right] \tag{6}$$

$$\left(F_{xf1} + F_{xf2}\right)\sin\delta + \left(F_{yf1} + F_{yf2}\right)\cos\delta + F_{yr1} + F_{yr2} = M\left[\dot{V}_y + \left(\gamma + \frac{d\beta_v}{dt}\right)V_x\right] \tag{7}$$

$$L_f(F_{yf1} + F_{yf2})\cos\delta - L_r(F_{yr1} + F_{yr2}) + \frac{L_w}{2}\left(F_{yf1} - F_{yf2}\right)\sin\delta + M_z = I_z\frac{d}{dt}\left(\gamma + \frac{d\beta_v}{dt}\right) \tag{8}$$

$$M_z = L_f(F_{xf1} + F_{xf2})\sin\delta + \frac{L_w}{2}(-F_{xf1} + F_{xf2})\cos\delta + \frac{L_w}{2}(-F_{xr1} + F_{xr2}) \tag{9}$$

where $\delta$ is the steer angle; $F_{xf1}$ and $F_{xf2}$ are longitudinal traction forces on the left and right front tires and these forces are assumed equal for $\delta = 0$; $F_{yf1}$ and $F_{yf2}$ are lateral traction forces on the left and right front tires; $F_{xr1}$ and $F_{xr2}$ are longitudinal traction forces on the left and right rear tires; $F_{yr1}$ and $F_{yr2}$ are lateral traction forces on the left and right rear tires; $\gamma$ is the yaw velocity; $\beta_v$ is the vehicle sideslip angle; $I_z$ is the mass moment of inertia in the yaw direction; and $M_z$ is the yaw moment for cornering. These equations were used to determine torque distributions when the vehicle is cornering.

**Table 1.** Vehicle Specifications.

| Vehicle Property | Symbol | Value |
|---|:---:|:---:|
| Frontal area of vehicle [m$^2$] | $A_f$ | 1.6 |
| Aerodynamic coefficient | $C_d$ | 0.28 |
| Cornering stiffness of the front tire [N/rad] | $C_f$ | 51,091 |
| Cornering stiffness of the rear tires [N/rad] | $C_r$ | 72,802 |
| Rolling resistance between tire and ground | $C_t$ | 0.01 |
| Height of equivalent aerodynamic point [m] | $h_a$ | 1 |
| Height of mass center [m] | $h_m$ | 0.56 |
| Yaw inertia of vehicle [kg·m$^2$] | $I_z$ | 1200 |
| Distance from mass center to front tire [m] | $L_f$ | 1.433 |
| Distance from mass center to rear tire [m] | $L_r$ | 1.067 |
| Distance between two rear wheels [m] | $L_w$ | 1.46 |
| Total mass of vehicle [kg] | $M$ | 1813 |
| Sprung mass of vehicle [kg] | $M_s$ | 1753 |
| Gear ratio | $n_g$ | 3 |
| Tire radius [m] | $r_t$ | 0.288 |

For simplicity, the hill climbing resistance, aerodynamic drag, and rolling resistance of the last terms in (1) are omitted, and the roll and pitch motions are neglected. However, the vehicle in CarSim for the real-time HIL simulation was modelled with self-contained yaw, roll, and pitch dynamics.

### 2.3. Tire Model

The CarSim tire lookup table that was obtained directly from the laboratory measurements was used to model the tire characteristics. The friction coefficient between the tire and road surface was chosen at 0.85, the longitudinal and lateral traction (or friction) forces on the tire were expressed as a function of normal force and tire slip ratio. Once the vehicle velocity, acceleration or deceleration, the normal forces $N_f$ and $N_r$ are known, the slip ratio of each wheel can be obtained. The front and rear wheel speeds $\omega_f$ and $\omega_r$ are therefore determined by the definition of tire slip ratio, as follows:

$$Acceleration: \ \omega = \frac{V_x}{r_t(1-\lambda)}, r_t\omega > V_x \tag{10}$$

$$Deceleration: \ \omega = \frac{V_x(1+\lambda)}{r_t}, r_t\omega < V_x \tag{11}$$

where the tire slip ratio $\lambda$ represents $\lambda_f$ and $\lambda_r$ that correspond to the speeds $\omega_f$ and $\omega_r$ of the front and rear wheels.

### 2.4. Transmission Model

After the wheel speeds and accelerations are obtained, the output torque $T_{mf}$ of the front traction motor, the output torque $T_{mr}$ provided by the right rear in-wheel motor, and the output torque $T_{ml}$ provided by the left rear in-wheel motor can be calculated under the following assumptions:

1. The rotor mass is so small compared with the vehicle mass that the rotational inertias of the three traction motors are neglected.
2. The viscous and Coulomb frictions of motors and differentials in the transmission are all neglected.

Therefore, when the vehicle accelerates:

$$Front\ wheels:\ \ T_{mf}n_g = \frac{2I_w\dot{\omega}_f}{n_g} + 2F_{xf}r_t \tag{12}$$

$$Rear\ wheels:\ T_{mi} = I_w\dot{\omega}_i + F_{xr}r_t, \quad i = r,\ l \tag{13}$$

where $n_g$ is the reduction ratio of gearbox; $I_w$ is the wheel inertia; $\omega_f$ is the speed of the front traction motor; $\omega_r$ and $\omega_l$ represent, respectively, the speed of the right and left rear in-wheel motors. In the steady state for a small steer angle and by neglecting frictions on wheel motors, the yaw moment Equation (9) can be simplified as

$$T_{mr} = T_{ml} + 2r_t M_z/L_w \tag{14}$$

where $L_w$ is the distance between two rear wheels. This equation will be used for determining the real-time torque distribution in the PSO process when the vehicle is cornering. For a straight-line driving, $M_z = 0$ and $T_{mr} = T_{ml}$.

*2.5. Battery State of Charge Model*

The SOC of each battery pack on the EV is estimated by the following equation:

$$SOC = SOC_i - \frac{\int_0^t I_b dt}{Q_b} \tag{15}$$

where $SOC_i$ is the initial state of SOC, $I_b$ is the battery current, and $Q_b$ is the battery maximum capacity. By a simple internal resistance model, the output power of battery is estimated by:

$$P_b = V_{oc}I_b - I_b^2 R_b \tag{16}$$

where $R_b$ is the internal resistance of the battery and was set at 0.17 $\Omega$ for all the battery packs in the MIL simulations in Section 4. The open circuit voltage (OCV) $V_{oc}$ is a function of SOC and is obtained from experiments for each battery pack. Figure 3 illustrates the OCV curve of the front battery pack. Accordingly, the battery current $I_b$ is easily calculated from (16).

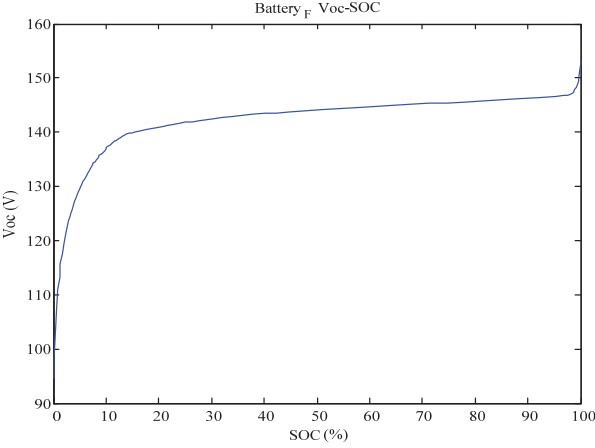

**Figure 3.** The open circuit voltage and state of charge curve of the front battery pack.

## 3. Torque and Battery Distribution Strategy

*3.1. Torque Distribution Strategy: Particle Swarm Optimization*

The total torque, $T_t$, used to accelerate the vehicle was

$$T_t = (F_x/r_t) = T_{mf}n_g + T_{mr} + T_{ml} \tag{17}$$

From (14), the corresponding yaw moment, $M_z$, in the steady state of a small steer angle was provided by the differential torque of the left and right in-wheel motors. The maximum and minimum torque ranges of the three traction motors were first determined on the basis of the TNE maps of the three traction motors, wheel angular speeds, SOC of the batteries, and current limits. PSO was then used to determine the best torque distribution for the three traction motors under the constraints of their operation ranges and Equations (14) and (17).

The PSO theory originated by observing the hunting behavior of a swarm of birds or fish. In the process of torque distribution, a particle is a point on a search space of TNE map. As shown in Figure 2a,b, three swarms of particles were initially distributed at random on the three TNE maps of the traction motors, and each swarm had N particles. Each particle had a position state, which was defined as the torque where the particle was located. The pedal command given by the EV driver was their common target at a specific vehicle speed.

Each particle at its initial position in the search space determined its best direction, and all the particles approached their common target with the minimal global effort.

$$\eta = \frac{T_{mf}\omega_{mf}}{\eta_{mf}\left(T_{mf}, \omega_{mf}\right)} + \frac{T_{ml}\omega_{ml}}{\eta_{ml}(T_{ml}, \omega_{ml})} + \frac{T_{mr}\omega_{mr}}{\eta_{mr}(T_{mr}, \omega_{mr})} \tag{18}$$

During the PSO process, the N particles were renewed through J generations and reached the target of minimal energy consumption in the end. After the least energy consumption converged in each generation, the particles were updated according to their own best and swarm best solutions as

$$\Delta T^{j+1}_{mf,i} = wT^{j+1}_{mf,i} + rand_1 \times c_{L1}\left(P^j_i - T^j_{mf,i}\right) + rand_2 \times c_{L2}\left(G^j - T^j_{mf,i}\right) \tag{19}$$

$$T^{j+1}_{mf,i} = T^j_{mf,i} + \Delta T^{j+1}_{mf,i} \tag{20}$$

where the sub-index i stands for the i[th] particle; j stands for the generation number; $c_{L1}$ and $c_{L2}$ are learning factors; $G$ represents the swarm's best known solution of all the particles; $P_i$ represents its own best known solution of the i[th] particle; $rand_1$ and $rand_2$ are random values between 0 and 1; and $w$ is the inertia weight. At the final generation, the output torques of the rear left and right motors were calculated:

$$T_{ml} = \frac{T_t - T_{mf}\,n_g}{2} - \frac{r_t M_z}{L_w} \tag{21}$$

$$T_{mr} = \frac{T_t - T_{mf}\,n_g}{2} + \frac{r_t M_z}{L_w} \tag{22}$$

Details of the original work of the real-time torque-distribution strategy by PSO for a pure EV with three traction motors were described in [16].

*3.2. Torque Distribution Strategy: Priority Torque Ratio in Front and Rear Motors*

For comparison, a priority torque ratio (PTR), $P_r$, can be assigned from 0 to 1 to the front motor with respect to the rear motors. For example, $P_r = 1$ means that the front motor takes the first priority for delivering the torque within its feasible range for driving the EV, while $P_r = 0$ expresses that the

rear motors have the top priority for delivering the torque over the front motor. Therefore, the torque given by the front motor is calculated as

$$T_{mf} = \left(T_{mf,max} - T_{mf,min}\right)P_r + T_{mf,min} \tag{23}$$

where $T_{mf,max}$ and $T_{mf,min}$ are, respectively, the upper and lower limits of the front motor at a certain speed. If $T_{mf}$ is calculated higher than or equal to the total torque $T_t$ required for acceleration, the front motor will provide $T_t$ as demanded. If $T_{mf}$ is calculated lower than $T_t$ even though $P_r = 1$, the rear motors must be responsible for delivering the rest portion of torque according to (21) and (22).

### 3.3. Battery Energy Consumption

The energy consumption and balance of the three battery packs were investigated when the torque distribution was executed by the PSO or PTR strategy. In this study, the urban driving cycle (UDC) of the New European driving cycle (NEDC) was used because of the limited motor speeds and battery voltage. Table 2 presents two simulation results for the energy consumption for the three battery packs after the EV drove (1) on a straight road and (2) clockwise on a circular path with a 100 m radius using the proposed PSO and PTR torque distribution strategies. For the cases of $P_r = 1$ and 0.75, the front traction motor had the highest priority by delivering power over the rear motors, and the rear battery restored more power from regenerative braking than that consumed for driving. This was an example of negative battery energy consumption.

**Table 2.** Energy consumption of battery packs.

| Strategy | | Battery Energy Consumption [Wh] | | | |
|---|---|---|---|---|---|
| | | Front | Rear Right | Rear Left | Total |
| **UDC on a Straight Road** | | | | | |
| | PSO | 223.2 | 37.76 | 37.76 | 298.72 |
| PTR | $P_r = 0$ | 44.00 | 158.4 | 158.4 | 361.16 |
| | $P_r = 0.25$ | 55.20 | 150.8 | 150.4 | 356.68 |
| | $P_r = 0.5$ | 91.60 | 123.2 | 123.2 | 338.44 |
| | $P_r = 0.75$ | 338.8 | −11.88 | −11.88 | 315.12 |
| | $P_r = 1$ | 351.2 | −19.72 | −19.72 | 311.92 |
| **UDC Clockwise along a Circular Path (Radius 100 m)** | | | | | |
| | PSO | 285.6 | 96.00 | 35.36 | 417.08 |
| PTR | $P_r = 0$ | 93.20 | 216.8 | 158.0 | 467.92 |
| | $P_r = 0.25$ | 113.2 | 205.2 | 146.0 | 464.28 |
| | $P_r = 0.5$ | 179.2 | 166.4 | 107.6 | 453.44 |
| | $P_r = 0.75$ | 429.2 | 32.32 | −14.28 | 447.36 |
| | $P_r = 1$ | 447.2 | 22.20 | −16.08 | 453.40 |

For the three battery packs, energy consumption was found to be unbalanced, thereby causing an unbalanced SOC. If any of the three battery packs was depleted, the torque distribution strategy would fail. This could cause a serious deterioration in vehicle maneuverability and stability.

### 3.4. Torque and Battery Distribution (TBD) Strategy

Figure 4 shows the flowchart for the torque and battery distribution (TBD) strategy. Before the EV started from rest, the SOC of the front, rear right, and rear left battery packs was respectively measured as $SOC_F$, $SOC_L$, and $SOC_R$. The SOC gap and SOC ratio were calculated:

1. SOC gap *(SOC$_g$)*: The SOC gap was defined as the difference between the SOC of the front battery pack and the lower SOC of the two rear battery packs. It was negative when the front battery had less power remaining than the rear batteries, and it was positive when the front battery had more

power than the rear batteries. In applications, a default value $SOC_g < -\kappa\%$ $(0 < \kappa < 2)$ can be assigned to determine the torque distribution mode (Figure 4).

2. SOC ratio ($SOC_r$): The SOC ratio was defined as the ratio of energy consumption in terms of the SOC between the front battery pack and the two rear battery packs. On the basis of the simulation of straight road driving under the PSO strategy, as indicated in Table 2, the $SOC_r$ converged to 2.94 for the long-term operation of the UDCs. In applications, a default value, $\rho$ ($< 2.94$), was assigned to determine the torque distribution mode under the TBD strategy.

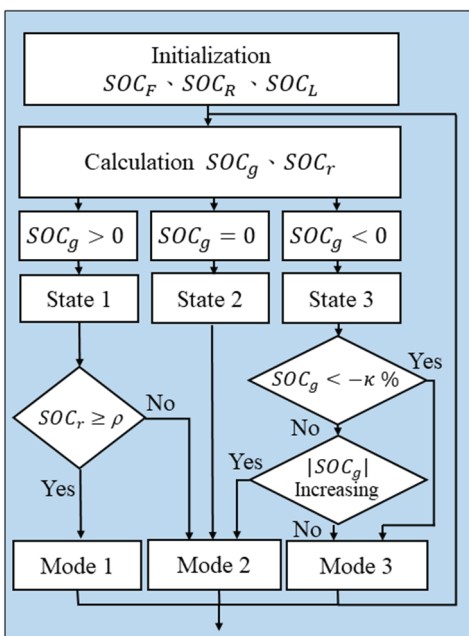

**Figure 4.** Flowchart of the torque and battery distribution strategy.

Through the use of $SOC_g$, the state of balance (SOB) of the three battery packs was investigated. The three states of battery balance are described:

1. State 1: The SOC of the front battery pack was higher than that of any of the rear battery packs.
2. State 2: The SOC of the front battery pack was equal to that of any of the two rear battery packs.
3. State 3: The SOC of the front battery pack was lower than that of the rear battery pack.

Three torque distribution modes were then determined by the SOC gap ($SOC_g$):

1. Mode 1: $P_r = 1$ was proposed when the front motor had the first priority for delivering the torque under the condition of $SOC_g > 0$ and $SOC_r \geq \rho$.

At Mode 1, the SOC of the front battery pack was much higher than that of the rear battery packs. When the $SOC_r$ was larger than 2.94, the $SOC_g$ could increase continuously, and the battery balance could worsen even though the PSO strategy was executed. It was better for the front battery pack to reach a balance between the front and rear batteries.

2. Mode 2: The PSO strategy was prescribed when there was not much difference in the SOC of the three battery packs under the following conditions: $SOC_g = 0$ or ($SOC_g > 0$ and $SOC_r < \rho$) or ($0 > SOC_g \geq -\kappa\%$ and $|SOC_g|$ was increasing).

Because the PSO strategy is superior to the PTR strategy for distributing the torque among the traction motors, it should be used whenever the difference in the SOC of the battery packs is negligible. For example, $SOC_g = 0$ is an ideal case in which all of the batteries are in balance, and the PSO strategy

can save more energy than the other PTR strategies when distributing the torque. With $SOC_g > 0$ and $SOC_r < \rho$, the amount of power stored by the front and rear batteries was similar and sufficient. It was also an appropriate situation for torque distribution under the PSO strategy.

When $0 > SOC_g \geq -\kappa\%$, there was not much difference in the energy storage of the battery packs, and it was still safe to execute PSO even though the SOC gap was increasing.

3.   Mode 3: $P_r = 0$ was proposed when the rear motors took top priority for delivering more the torque than the front motor under the following conditions: $SOC_g < -\kappa\%$ or ($0 > SOC_g > -\kappa\%$ and $|SOC_g|$ was decreasing).

At Mode 3, the SOC of the rear battery packs was much higher than that of the front battery. The consumption in the rear batteries had top priority so that the balance in the batteries could be restored. After the $SOC_g$ was restored within $[0, -\kappa\%]$, Mode 3 remained in operation because the SOC gap continued to decrease until the best balance state 2 was achieved. This avoided frequent shifts between Modes 2 and 3.

## 4. Experiments

### 4.1. Model-in-the-Loop Simulations

In Figure 5, the TBD strategy was simulated on a model-in-the-loop (MIL) platform. MATLAB Simulink was applied to model the TBD strategy, battery model, driver model, slip ratio control (SRC), and direct yaw moment control (DYC), while CarSim provided the vehicle dynamics. The driver model simulated human driver behavior by a proportional-integral (PI) controller. The SRC was responsible for stabilizing vehicle motion through the tractive control system (TCS) during acceleration and the anti-lock brake system (ABS) during deceleration. The TBD strategy was performed after vehicle safety was confirmed. Because of the limited motor speeds and battery voltages, only the UDC part of NEDC was used for MIL simulations.

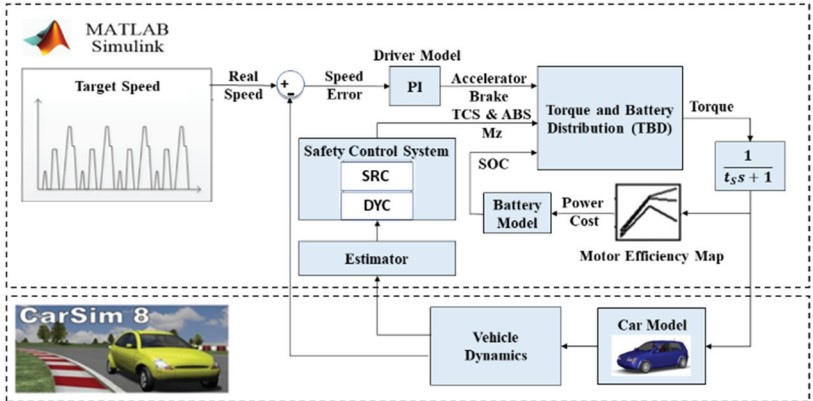

**Figure 5.** Simulation block diagram of torque and battery distribution strategy.

Both the straight and circular road tests were simulated. Figure 6a shows the SOC records of the three battery packs during the TBD process for the EV driving a clockwise cornering on a circular path of radius 100 m for 3.5 UDCs of the NEDC. The indices $\rho$ and $\kappa$ were assigned at 1.0056 and 1, respectively.

At the beginning, the SOC of the front battery (91%) was higher than that of both rear batteries (90%). In addition, $SOC_r > 1.0056$, the torque distribution of Mode 1, was executed until the $SOC_r$ reduced to 1.0056 at approximately 170 s, where the PSO of Mode 2 was executed for torque distribution.

The torque distribution mode shifted from Mode 2 to Mode 3 when the $SOC_g$ was less than $-1\%$ at approximately 1375 s. The torque distribution mode shifted back to Mode 2 about 1575 s when the SOC gap was reduced. The SOC of the front and rear batteries remained within a SOB by shifting

the torque distribution mode during the driving cycle. The torque distribution histories from three traction motors are presented in Figure 6b.

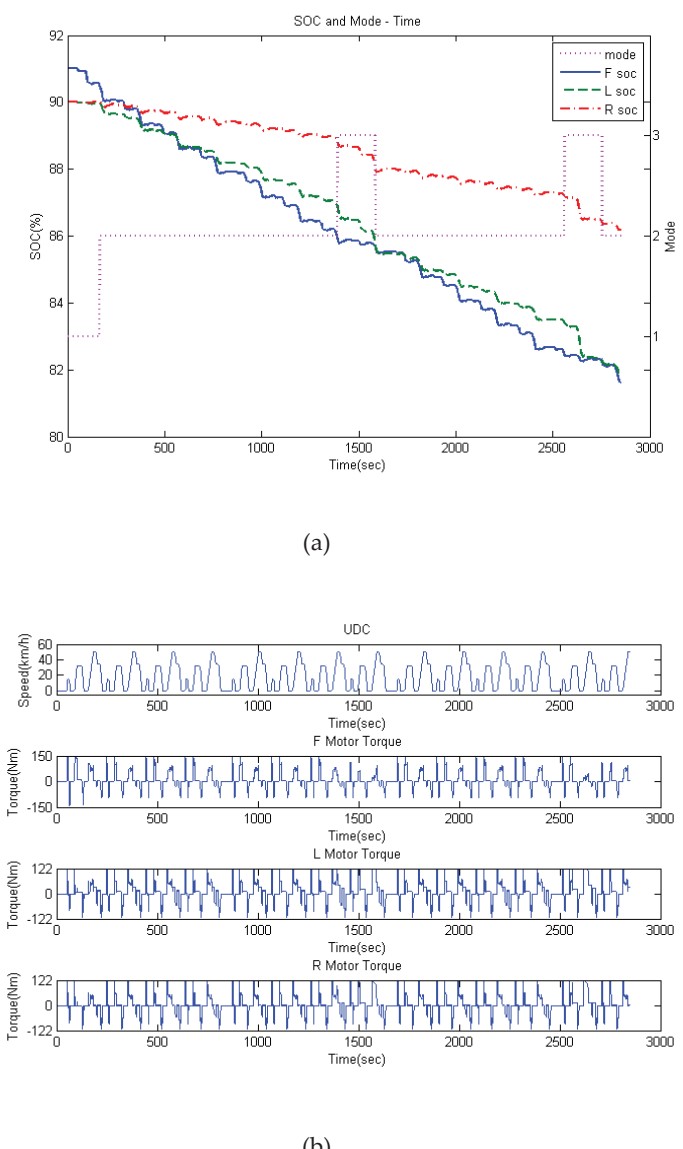

(a)

(b)

**Figure 6.** (**a**) The state of charge records for the three battery packs and (**b**) torque distribution from the front, rear left, and rear right traction motors during the torque and battery distribution process for the electric vehicle driving on a circular path of radius 100 m and using the urban driving cycle of the New European Driving Cycle.

It was also interesting to compare the differences of the energy economy of the proposed TBD strategy by having the battery energy storage in balance and using other torque distribution strategies without balancing the batteries. In these simulations, the initial SOC of each of the three battery packs was 90%, and their corresponding amounts of energy are 10.45, 5.2, and 5.2 kWh for the front, rear right, and rear left battery packs, respectively. The EV stopped when any of the batteries was depleted. It was found that both the rear batteries were exhausted soon for $P_r = 0$, 0.25, and 0.5 when the rear motors took higher priority for delivering more torque than the front motor; while the front battery was depleted soon for $P_r = 0.75$ and 1 when the front motor had the first priority for delivering the torque to the EV.

Energy consumption efficiency was defined as the ratio between the total energy consumption and the initial battery energy. The energy consumption rate was defined as the total energy consumption per travel distance.

Table 3 shows the energy consumption results for both the circular and straight road simulations. For clockwise cornering on a circular path of radius 100 m during the UDCs, the proposed TBD strategy had a travel distance: 142.6 km, which was 7.7% higher than the PSO strategy without the balancing of the SOC of the battery packs. The torque distribution strategy under PSO without battery balancing had a better energy consumption rate at 104.5 Wh/km than the TBD strategy, but the front battery was delpeted after 132.4 km, and the energy consumption efficiency was 73.6%. The rear battery packs had 26.4% energy remaining when the EV stopped.

**Table 3.** Torque and battery distribution strategy results.

| Strategy | Energy Consumption (kWh) | Travel Distance (km) | Energy Consumption Rate (Wh/km) | Energy Consumption Efficiency (%) |
|---|---|---|---|---|
| Clockwise UDC along a Circular Path (Radius 100 m) | | | | |
| TBD | 16.52 | 142.6 | 115.9 | 88.0 |
| PSO | 13.89 | 132.4 | 104.9 | 73.5 |
| $P_r = 0$ | 10.11 | 85.9 | 117.7 | 53.3 |
| $P_r = 0.25$ | 11.10 | 90.6 | 122.5 | 58.6 |
| $P_r = 0.5$ | 12.76 | 111.9 | 114.1 | 67.4 |
| $P_r = 0.75$ | 9.91 | 88.2 | 112.4 | 52.2 |
| $P_r = 1$ | 9.64 | 84.5 | 114.0 | 50.8 |
| UDC on a Straight Road | | | | |
| TBD | 18.75 | 214.4 | 87.4 | 99.9 |
| PSO | 12.59 | 167.6 | 75.1 | 67.1 |
| $P_r = 0$ | 10.66 | 117.4 | 90.8 | 56.8 |
| $P_r = 0.25$ | 11.08 | 123.5 | 89.7 | 59.0 |
| $P_r = 0.5$ | 12.85 | 151.0 | 85.1 | 68.5 |
| $P_r = 0.75$ | 8.74 | 110.3 | 79.3 | 46.6 |
| $P_r = 1$ | 8.35 | 106.4 | 78.4 | 44.5 |

On the straight road, the torque distribution strategy under PSO without battery balancing exhibited the best energy consumption rate: 75.21 Wh/km. However, the front battery was depleted after 167.6 km, and energy consumption efficiency was 67.5%. Thus, the rear battery packs had only 32.5% energy remaining when the EV stopped. The proposed TBD strategy of partly using torque distribution by PSO had the highest driving range: 214.4 km, i.e., approximately 248 UDCs. This was attributed to the SOC of the three battery packs being kept in balance to avoid unexpected battery depletion. Therefore, approximately 28% more driving range was extended by the TBD strategy than by the PSO strategy without battery balancing.

It was also found in the simulation that the battery energy was fully utilized by the TBD strategy for the EV on a straight road of UDC. The 99.9% energy consumption efficiency for the TBD strategy was calculated in Table 4. The energy of three battery packs can be effectively distributed and utilized to extend the driving range, when the SOC gap of the three battery packs remains within a prescribed limit during vehicle operation.

**Table 4.** Energy consumption efficiency for the EV on a straight road of UDC by the TBD strategy.

| Battery | Front | Rear Left | Rear Right |
|---|---|---|---|
| Energy capacity (kWh) @ 100% SOC | 10.45 | 5.2 | 5.2 |
| Initial SOC (%) | 90 | 90 | 90 |
| Final SOC (%) | 0 | 1.75 | 1.75 |
| Initial battery energy (kWh) | $A = (10.45)(0.9) + (5.2)(0.9)(2) = 18.77$ | | |
| Total energy consumption (kWh) | $B = (10.45)(0.9 - 0) + (5.2)(0.9 - 0.0175)(2) = 18.75$ | | |
| Energy consumption efficiency | $A/B = 18.75/18.77 = 99.89\%$ | | |

Other strategies with a PTR, $P_r$ from 0 to 1, presented lower travel distances and less energy consumption efficiency than observed for the proposed TBD strategy.

### 4.2. Hardware-in-the-Loop Simulations

Figure 7 presents the architecture of hardware-in-the-loop experiment. A Mitsubishi Colt-Plus was retrofitted with a 15-kW radial-flux PMSM and a 144-V battery pack for front wheels and two 7-kW axial-flux PMSMs and two 72-V battery packs for rear wheels. This EV was set up on a Horiba MAHA-AIP ECDM-48 emission chassis dynamometer. The maximum test speed was 200 km/h, the maximum power absorbing was 150 kW at 100 km/h, the maximum tractive force was 5400 N for light duty and 6750 N for heavy duty, and the maximum vehicle inertia simulation was 4540 kg for 4-wheel drives.

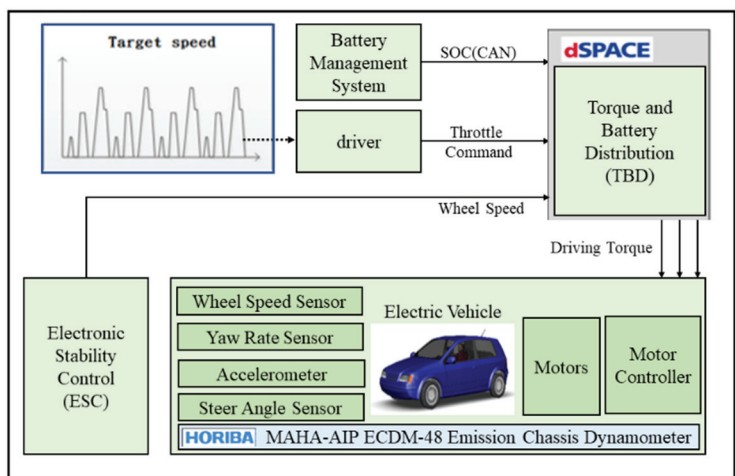

**Figure 7.** Architecture of the hardware-in-the-loop experiment.

In experiments, the battery SOC information and throttle command were received by a real-time rapid control prototyping unit dSPACE MicroAutoBox II, in which the TBD strategy was built to determine the torque command for each motor. This MicroAutoBox took a role of vehicle control unit with an 800-MHz processor, 18-MB main memory, 16-MB flash memory, and dual CAN interfaces.

In the HIL experiment, the vehicle followed the complete NEDC but the maximum speed at EUDC was restricted at 40 km/h in the HIL experiment. The initial SOCs of the front, rear left, and rear right batteries were 94.8%, 93.6%, and 95.5%, respectively. In order to make the experiment efficient, driving modes were shifted at $SOC_g = -\kappa\% = -1\%$ and $SOC_r = \rho = 1.006$, according to the flowchart of the TBD strategy in Figure 4.

Figure 8 illustrates the time history of mode, state, $SOC_r$, and $SOC_g$ and the corresponding torque distribution histories of the front, rear left, and rear right motors during the TBD process in the hardware-in-the-loop experiment. During the first 95 s, the SOC of the front battery pack was higher than the SOC of anyone of the rear battery packs. Therefore, State 1 ($SOC_g > 0$) was identified and $SOC_r$ was larger than 1.006, the front motor took the first priority of delivering torque and Mode 1 ($P_r = 1$) was executed. Between 95 and 200 s, it was still at State 1 ($SOC_g > 0$) but $SOC_r$ was less than 1.006, Mode 2 was executed with the PSO strategy.

Between 200 and 300 s, State 2 ($SOC_g = 0$) was identified, i.e., the SOC of the front battery pack was equal to any one of the two rear battery packs. Mode 2 with the PSO strategy was working. Between 300 and 485 s, the SOC of the front battery pack was lower than the SOC of the rear battery packs, and State 3 ($SOC_g < 0$) was identified. Because $0 > SOC_g \geq -1\%$ and $|SOC_g|$ was increasing, Mode 2 remained.

When $SOC_g$ was less than −1% at State 3 between 485 and 585 s, the rear battery packs and motors started to take their top priority for delivering more power than the front battery and motor, and Mode 3 ($P_r = 0$) was executed. Between 585 and 680 s, the SOC gap was between −1% and 0%, but $|SOC_g|$ was decreasing, Mode 3 remained.

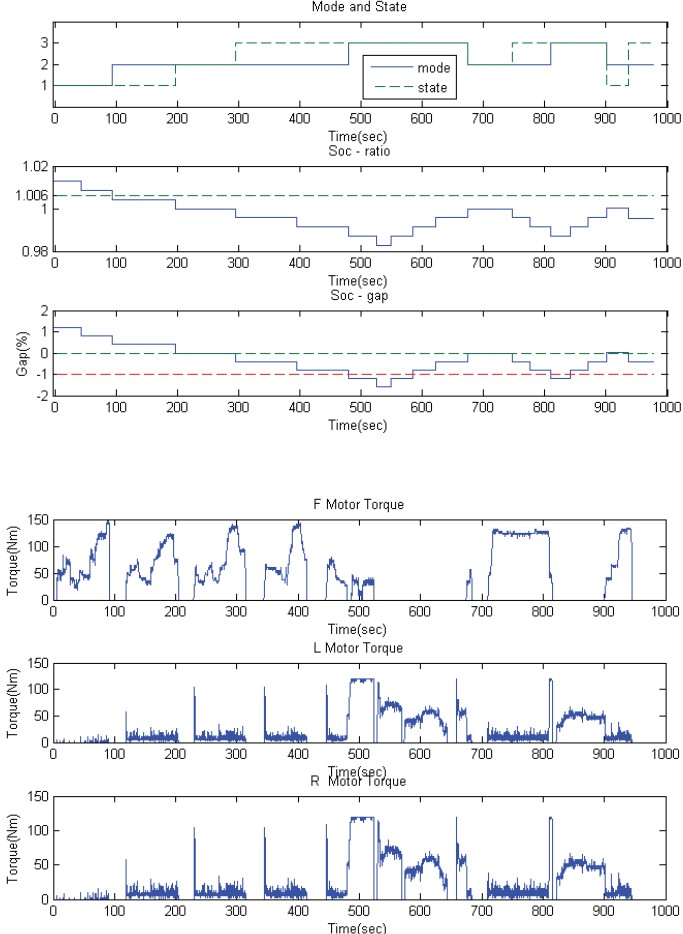

**Figure 8.** Histories of the mode (solid line), state (dashed line), state of charge (SOC) ratio, SOC gap and the torque distributions of the front, rear left, and rear right motors during torque and battery distribution process for the electric vehicle in the hardware-in-the-loop experiment.

The corresponding SOCs of the three battery packs is shown in Figure 9. In the first 500 s, the front battery provided all power to the vehicle. After 500 s, the rear and front batteries powered the vehicle alternatively, so that the SOC gap would always remain in 1% as expected.

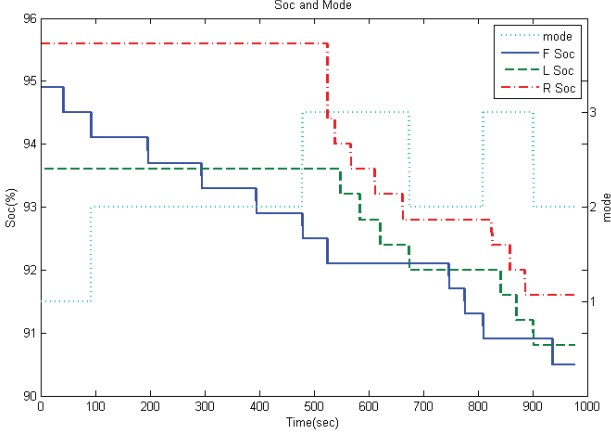

**Figure 9.** Histories of mode and the states of charge of the front, rear left, and rear right battery packs during torque and battery distribution process for the electric vehicle in the hardware-in-the-loop experiment.

### 4.3. Road Tests

The road test was executed on the Industrial Technology Research Institute (ITRI) campus. The total driving distance was about 5.4 km, during which the road slope varied and the maximum speed was 30 km/h. The driving curve is shown in Figure 10. The initial SOCs of the front, rear left, and rear right batteries were 72.8%, 70%, and 71.2%, respectively. Driving modes were shifted at $SOC_g = -\kappa\% = -1\%$ and $SOC_r = \rho = 1.02$, according to the flowchart of the TBD strategy in Figure 4. Figure 11 illustrates the time history of mode, state, $SOC_r$, $SOC_g$, and the corresponding torque distributions.

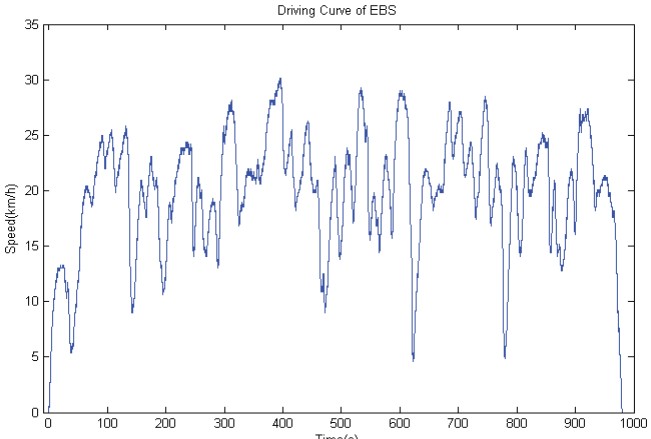

**Figure 10.** Driving speed curve for the torque and battery distribution strategy in the road test on the campus of Industrial Technology Research Institute.

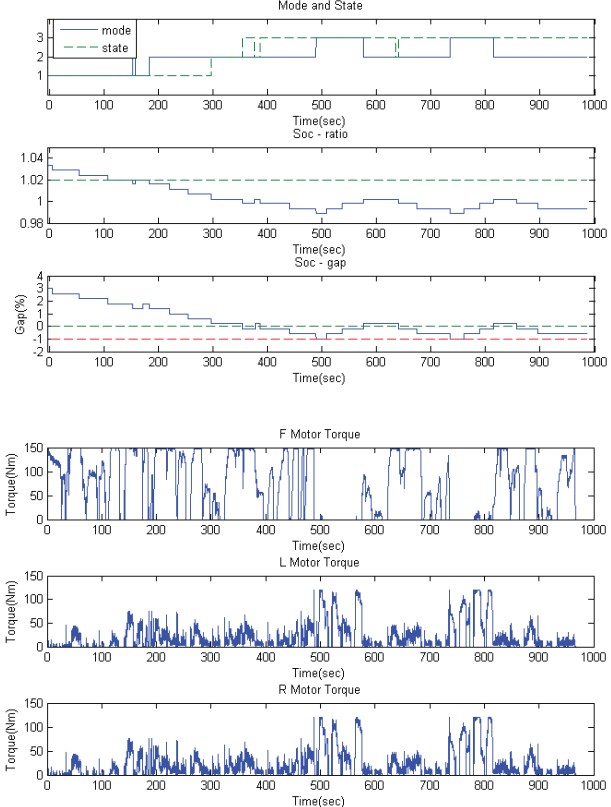

**Figure 11.** Histories of the mode (solid line), state (dashed line), state of charge (SOC) ratio, SOC gap and the torque distributions of the front, rear left, and rear right motors during torque and battery distribution process for the electric vehicle in the road test.

Similar to the result of HIL test, the front battery pack took the first priority to supply power to drive the EV at Mode 1 ($P_r = 1$) during the first 180 s. Because the EV moved upslope approximately at 150 s, two rear motors delivered extra torque. Between 180 and 480 s, Mode 2 with the PSO strategy worked when the SOC of the front battery was lower than that of the rear battery packs. The front motor still provided the major torque until the SOC gap was less than −1% ($SOC_g < -1\%$) while Mode 3 ($P_r = 0$) was executed. Then, Modes 2 and 3 shifted alternatively. The SOC difference of the three battery packs was finally kept within 1%, as shown in Figure 12.

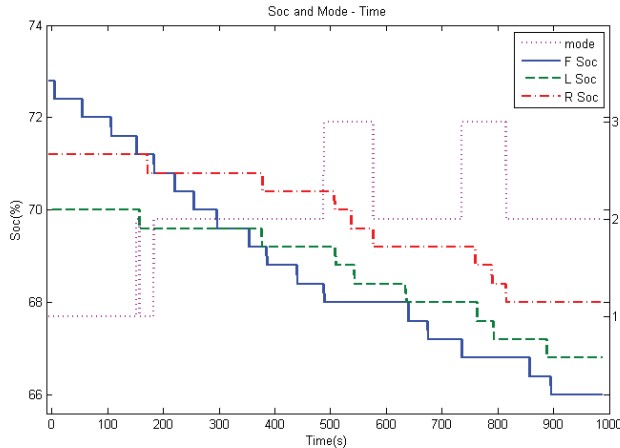

**Figure 12.** The variation of SOC of the three battery packs in the road test.

The two strategies with a PTR, $P_r = 0$ and $P_r = 1$, were also executed in the road test. The total energy consumption and the consumption of each battery pack are given in Table 5. It shows that the energy consumption rate of the TBD strategy was lower than those of $P_r = 0$ and $P_r = 1$. It means that with the same battery energy, the EV has a longer driving range by the proposed TBD strategy than that by other PTR strategies. For example, the TBD strategy extended 11% more driving range than the PTR strategy when $P_r = 0$, and the TBD strategy extended 23.5% more driving range than the PTR strategy when $P_r = 1$.

**Table 5.** Battery energy consumption in road tests.

| Strategy | Battery Energy Consumption [Wh] | | | Driving Range (m) | Energy Consumption Rate [Wh/km] |
|---|---|---|---|---|---|
| | F | RL | RR | | |
| TBD | 797 | 149 | 149 | 5464 | 200 |
| $P_r = 0$ | 179 | 515 | 510 | 5424 | 222 |
| $P_r = 1$ | 1318 | 14 | 18 | 5470 | 247 |

F: Front battery, RL: Rear left battery, RR: Rear right battery.

## 5. Conclusions

A novel TBD strategy has been proposed for the EV with three independent traction motors and battery packs. Upon acceleration of the EV, the demanded torque was provided by all three traction motors together at their highest efficiency under the PSO strategy for saving battery energy. Simultaneously, the SOC of the three battery packs had to be kept in balance to avoid any unexpected battery depletion and to improve the EV's driving range. Thus, a combination of PSO and the PTR for torque distribution strategy was applied to compromise between energy saving and energy balance. On the basis of the model-in-the-loop simulations, the proposed TBD strategy shows better travel distance and the higher energy consumption efficiency than a pure PSO method or PTR strategies. Similar results were also proved on a real vehicle for hardware-in-the-loop experiments on dynamometer and road tests. The road test proved that the TBD strategy extended 11% and 23.5%

more driving range than other PTR strategies, when $P_r = 0$ and $P_r = 1$, respectively. The proposed TBD strategy is promising for extending the driving range of an EV with multiple traction motors and battery packs with an improved energy consumption efficiency.

**Author Contributions:** Y.-H.T. and Y.-P.Y. conceived and design the experiments; Y.-H.T. performed the experiments and analyzed the data; Y.-P.Y. wrote the paper. All authors have read and agreed to the published version of the manuscript.

**Funding:** This research was funded by the Ministry of Science and Technology of Taiwan, Republic of China under contract MOST 106-2221-E-002-064.

**Acknowledgments:** The authors acknowledge the financial support of the Ministry of Science and Technology of Taiwan, Republic of China under contract MOST 106-2221-E-002-064.

**Conflicts of Interest:** The authors declare no conflicts of interest.

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
