# Peer review of "Torque and Battery Distribution Strategy for Saving Energy of an Electric Vehicle with Three Traction Motors"

_applsci, doi:10.3390/app10082653_

Round 1

Reviewer 1 Report

Comments and Suggestions for Authors:

Authors did not explain clearly why using three battery packs. Apart from an arguable redundancy, I don't see any other advantage. You need three chargers, more complex cooling and heating systems (if placed apart) and three BMS systems. Until these aspects are not clarified, the article lacks for a technical challenge that should be solved.

Reviewer 2 Report

the paper is much interesting and the argument is much actual. also the simulation of a driving cycle in the circular line is interesting.

i suggest to use the new WLTC driving cycle instead of the NEDC that is less realistic.

There are some other questions:

  1. Multiple motors often are used in high specific power vehicles ,why did you use 3 motors in a such little vehicle.
  2. Why 3 battery packs instead of one? probably only one pack  or two (one for axys)  is a simpler and more efficient choice.
  3. Are in equation 16 the  battery voltage Vdc  and the internal resistance Rb constant during your simulations? if not could you add more details about the battery behaviour with SOC?

Round 2

Reviewer 1 Report

Acceptable in present form.